# Consumption of Breast Milk Is Associated with Decreased Prevalence of Autism in Fragile X Syndrome

**DOI:** 10.3390/nu13061785

**Published:** 2021-05-24

**Authors:** Cara J. Westmark

**Affiliations:** 1Department of Neurology, University of Wisconsin, Madison, WI 53706, USA; westmark@wisc.edu; Tel.: +1-608-262-9730; 2Molecular & Environmental Toxicology Center, University of Wisconsin, Madison, WI 53706, USA

**Keywords:** allergies, autism, breast milk, fragile X syndrome, gastrointestinal issues, seizures

## Abstract

Breastfeeding is associated with numerous health benefits, but early life nutrition has not been specifically studied in the neurodevelopmental disorder fragile X syndrome (FXS). Herein, I evaluate associations between the consumption of breast milk during infancy and the prevalence of autism, allergies, diabetes, gastrointestinal (GI) problems and seizures in FXS. The study design was a retrospective survey of families enrolled in the Fragile X Online Registry and Accessible Research Database (FORWARD). There was a 1.7-fold reduction in the prevalence of autism in FXS participants who were fed breast milk for 12 months or longer. There were strong negative correlations between increased time the infant was fed breast milk and the prevalence of autism and seizures and moderate negative correlations with the prevalence of GI problems and allergies. However, participants reporting GI problems or allergies commenced these comorbidities significantly earlier than those not fed breast milk. Parsing the data by sex indicated that males exclusively fed breast milk exhibited decreased prevalence of GI problems and allergies. These data suggest that long-term or exclusive use of breast milk is associated with reduced prevalence of key comorbidities in FXS, although breast milk is associated with the earlier development of GI problems and allergies.

## 1. Introduction

The American Academy of Pediatrics recommends exclusive breastfeeding for the first 6 months of life and continued breastfeeding with the introduction of complementary foods [1]. Breastfeeding is the ideal nutrition for babies and is associated with numerous health benefits, for example, reduced incidence of autism spectrum disorder (ASD), attention-deficit/hyperactivity (ADHD) disorder, obesity, sudden infant death syndrome (SIDS), infections and atopic disease. Specifically, children with ASD, whether by clinical diagnosis or self-report, are significantly less likely to have been breastfed [2]. Data from the Centers for Disease Control and Prevention (CDC) regarding several million children in the United States indicate a highly significant inverse relationship between ADHD and exclusive 3-month and 6-month breastfeeding [3]. Meta-analysis with a total over 226,000 participants indicates that breastfeeding is associated with significantly reduced risk of obesity in children with a dose response effect dependent on duration of breastfeeding [4]. Exclusive breastfeeding for less than 4 months is associated with increased risk of chest infection and diarrhea [5]. Any breastfeeding greater than 2 months is protective against sudden infant death syndrome (SIDS) with greater protection observed with longer duration of breastfeeding [6]. And exclusive breastfeeding for 3–4 months decreases the incidence of eczema while longer breastfeeding is protective against wheezing in the first 2 years of life [7].

Despite extensive knowledge on the health benefits of breastmilk for typically developing infants, there is a dearth of knowledge regarding the effects of breast milk on neurodevelopmental disabilities such as fragile X syndrome (FXS). FXS results from a trinucleotide repeat expansion in the 5′-untranslated region (UTR) of the fragile X mental retardation gene (*FMR1*) gene located on the X-chromosome [8]. The repeat expansion becomes hypermethylated and prevents production of fragile X mental retardation protein (FMRP) [9], which is a multifunctional messenger RNA (mRNA) binding protein with diverse cellular functions [10]. FXS affects about 1 in 5000 males and 1 in 4000–8000 females [11], and causes intellectual disability, autism and seizures [12].

Diet significantly impacts seizure severity in a mouse model of FXS (*Fmr1^KO^* mice). Specifically, a soy-based rodent chow increases audiogenic-induced seizures in *Fmr1^KO^* mice [13]. Retrospective analysis of medical record data from the Simons Foundation Autism Research Initiative (SFARI) indicates that consumption of soy-based infant formula is associated with increased febrile seizures, simple partial seizures, epilepsy comorbidity and autism phenotypes in a population of children with autism [14,15]. These data elicit the hypotheses that the consumption of soy-based infant formula could be associated with more severe FXS phenotypes [16,17] and that breastfeeding could be neuroprotective. To address the first hypothesis, a retrospective survey study, using the Fragile X Online Registry with Accessible Database (FORWARD [18]) as a sampling frame, indicates that soy-based infant formula is associated with statistically increased comorbidity of autism (1.5-fold), gastrointestinal (GI) problems (1.9-fold) and allergies (1.7-fold) [19]. The FORWARD nutrition survey also included data collection on the prevalence and duration of feeding with breast milk and reasons for commencement and cessation of breast milk. Herein, those results are correlated with the prevalence and onset of comorbid disorders in FXS.

## 2. Methods and Subjects

### 2.1. Study Design

A national registry of FXS families maintained by FORWARD was utilized to conduct a retrospective survey study evaluating associations between early childhood feeding practices and the severity of common FXS phenotypes. A detailed description of the study design, questionnaire, study population, recruitment success, and data collection have been previously described [19]. Briefly, the 55-point questionnaire, Fragile X Syndrome Nutrition Study, assessed demographics, infant feeding practices, frequency and severity of seizures, cognitive ability, autistic behaviors and comorbid diagnoses in an FXS population by parental survey. Eligibility for study participation included full-mutation FXS status, enrollment in FORWARD, and a completed Clinical Report form with seizure history. Exclusion criteria included lack of clinic or caregiver agreement to participate in the research study. All eligible participants with FXS and a seizure history and 4-fold more participants with FXS and no seizure history were contacted for participation. The primary predictor variable examined was the type of infant milk (casein, soy, breast) consumed.

### 2.2. Data Collection

Data collection occurred in two phases. Phase I occurred from 27 August 2019 through 6 November 2019 and consisted of sending prepared sample packets to 10 FXS clinics across the United States and the clinics mailing the packets to eligible participants. The clinics included: Children’s Hospital Boston, Children’s Hospital Colorado, Children’s National Health System, Cincinnati Children’s Hospital Medical System, Emory University School of Medicine, Geisinger Medical Clinic, New York Institute for Basic Research, Rush University Medical Center, University of California Davis Health System, and University of Minnesota. The participation response rate was 21%. Phase 2 occurred from 19 December 2019 through 7 February 2020 and consisted of mailing the questionnaire and reminders to participants who agreed to participate in the study. The participation response rate was 89% [19]. 

### 2.3. Statistics

This manuscript addresses the hypothesis that breast milk is associated with decreased comorbidities in FXS. Data were analyzed in accordance with STROBE guidelines (Appendix A). Variables of interest included caregiver reported data from survey questions (Q) (Appendix A) for incidence of autism, food allergies and diabetes (Q1); history of GI problems (Q2), age when GI problems started (Q3); history of seizures (Q4); age of first seizure (Q7); history of allergies (Q15); age of allergy onset (Q16); allergens (Q17); fed breast milk (Q18); age began feeding breast milk (Q19), reasons for feeding breast milk (Q20); age stopped being fed breast milk (Q21); reasons stopped feeding with breast milk (Q22); sex (Q48); and age (Q50 & Q55). Analysis of seizure type (Q5) was not assessed herein because caregiver reports of seizure type did not match clinical records [19]. Data were not corrected for infant length (Q51) or weight (Q52) measurements, which were identical between subjects classified as autistic by Q1a [20.2 (0.23) inches and 7.57 (0.13) pounds] and not autistic [19.8 (0.19) inches and 7.50 (0.14) pounds (*p* = 0.18 and 0.70, respectively). Percentages, means, standard error of the means (SEM), odds ratios (OR) and 95% confidence intervals (CI) were computed to describe the cohorts. Fisher exact test (if less than 5 outcomes per cell) or Pearson’s uncorrected chi-square tests were used to examine the null hypotheses that the prevalence of comorbidities in FXS are the same in infants fed breast milk or not. Student’s *t*-tests were used to compare the means of two cohorts. Statistical significance was defined as *p* < 0.05. A Bonferroni correction was not applied for multiple comparisons except where indicated. The numbers of participants for each comparison are reported in the corresponding tables. Participants were excluded from analyses when there was missing data on a variable of interest.

## 3. Results

### 3.1. Study Population

The study population was previously described and included 199 full mutation participants with FXS [19]. Participants were predominantly Caucasian and 73% male as reported by caregivers. The average age of participants was 18 years (SEM 0.74 years; based on data provided for *n* = 198). In terms of infant feeding, respondents (194 out of 199; 97%) provided a Yes or No answer to Question #18 of the Fragile X Syndrome Nutrition Study, “During the first year of life, was your child with Fragile X Syndrome fed any breast milk?”, with 73% indicating the use of breast milk, which agrees with Centers for Disease Control and Prevention (CDC) data indicating national averages of 77–84% between 2010–2017 for breastfeeding in the general population [20]. Of breastfeeding respondents, (*n* = 141 out of 142: 99%) provided start and end dates for breastfeeding with only a small percentage discontinuing breastfeeding in 2 weeks or less and a large percentage continuing breastfeeding 3 months or longer (Table 1). Of breastfeeding respondents, a large proportion (29%) reported exclusive use of breast milk (i.e., no use of cow, soy or specialty infant formulas) with the period of exclusive breast feeding ranging from 2–30 months [Mean 13.4 months (0.91); *n* = 43]. There were no statistically significant differences between males and females in terms of breast milk use although a larger study population would discern if the trends for increased breastfeeding in males from 3–6 months of age are significant (Table 1). Compared to 2010–2017 CDC national averages of 48–58% for breastfeeding at 6 months of age, the FORWARD participants had 52% reported breastfeeding at 6 months of age for the total study population (*n* = 199).

### 3.2. Comorbid Conditions as a Function of Infant Feeding with Breast Milk

There were no statistically significant differences in the prevalence of autism, food allergies, diabetes, GI problems, seizures or allergies between the cohort reporting being fed any breast milk during the first year of life and the cohort that was never breast fed (Table 2). Accounting for the length of time participants were fed breast milk (at least 3, 6 or 12 months) resulted in a statistically significant 1.7-fold reduction in autism (from 53% to 32%) with 12-months feeding with breast milk (Table 3). The range of autism prevalence is within cited rates of 15–67% of comorbid autism in FXS [21,22]. Caregiver-reported incidence of autism (Q1a) agreed with reported autistic behaviors (Q40–Q46) (Table 4). There was a strong positive correlation between increased time fed breast milk and prevalence of no comorbidities; strong negative correlations between increased time fed breast milk and autism and seizures; and moderate negative correlations between increased time fed breast milk and GI problems and allergies (Table 5). Infants with caregiver-reported autism were 2-fold less likely to be fed breast milk at 12 months of age (Appendix A). Considering only the participants that reported exclusive use of breast milk, there was a 1.9-fold reduced prevalence of autism in the breast milk cohort (*p* = 0.019) (Table 6). These findings corroborate extensive literature that shows that breast milk is associated with improved development. In contrast, there were significant associations between the use of soy-based infant formula and increased prevalence of autism (1.5-fold), GI problems (1.9-fold) and allergies (1.7-fold) [19].

### 3.3. Caregiver-Reported Reasons for Starting & Stopping Breastfeeding

In response to Question #20 of the Fragile X Syndrome Nutrition Study, “Some examples of why people might choose to feed their child breast milk include doctor recommendations, family recommendations, or a belief that it is healthiest for the baby. Why was your child fed breast milk?” the most common response was that it was healthiest for the baby (Table 7). In response to Question #22, “Some examples of why people might choose to stop feeding their child breast milk include being painful for the mother, or the baby was not getting enough milk. What was the reason your child stopped being fed breast milk?”, the most common reasons for stopping breastfeeding were milk production and the age of the child followed by the mother needed to wean the child usually for work-related issues, it was painful to the mother, and/or the child had a health-related problem that interfered with breastfeeding (Table 7).

### 3.4. Seizures in the Study Population

The primary hypothesis under evaluation in our Fragile X Syndrome Nutrition Study was that the prevalence of seizures in FXS would be higher in infants fed soy-based infant formula [19]. Conversely, breast milk would be expected to be protective against seizures. I did not find a statistically significant difference in seizure prevalence as a function of breast milk with the small study size, albeit there was a trend for decreased seizures with breast milk use for 12 months or longer (*p* = 0.064) (Table 3). There was also a trend for a younger age of the first seizure in the breast milk cohort (*p* = 0.23) (Appendix A).

### 3.5. Gastrointestinal Problems in the Study Population

Respondents provided data regarding the use of breast milk and the start of GI problems in their children with FXS. A larger percentage of participants commenced GI problems within 3 years of age when fed breast milk (Table 8). The average age of commencing GI problems was 3.6 years earlier for the breast milk cohort. The average age of commencing GI problems in the exclusive breast milk cohort was 16 months (*n* = 8, SEM = 5.9). Those 8 participants had a 38% prevalence of autism, 29% food allergies and 57% allergies. These data suggest that while long-term and exclusive use of breast milk are protective against autism, breast milk is associated with the earlier development of GI problems in participants with FXS. For comparison, participants fed soy-based infant formula had an average age of commencing GI problems of 7.8 (2.7) months versus no soy at 52 (15) months (*p* = 0.031) [19]. Participants exclusively fed cow milk-based formula exhibited a 50% prevalence of GI problems (*n* = 16) with an average age of commencing GI problems at 11 (3.1) years. Thus, both soy-based infant formula and breast milk are associated with an earlier age of GI problems in FXS.

### 3.6. Allergies in the Study Population

Respondents reported data regarding the age of commencement of allergies in their children with FXS. Allergies commenced an average of 38 months earlier with a 1.7-fold increase in prevalence within 3 years of age in FXS participants fed breast milk (Table 8). All participants reporting the timing of allergies exhibited symptoms by 13 years of age. Participants reporting allergies within 2 weeks of age included milk or lactose as an allergen. The average age of allergies in the participants fed breast milk only cohort was 24 months (*n* = 12; SEM 5.3) with all participants reporting allergies within 6 years of age. In comparison, participants fed soy-based infant formula reported allergies commencing at 28 (6.7) months versus 50 (7.2) months fed no soy (*p* = 0.06) with statistically significant increases in prevalence within 2 weeks and 1 year of age (*p* < 0.05) [19]. Participants exclusively fed cow milk formula (*n* = 16) had a 63% prevalence of allergies with age data available for 9 participants who reported an average onset of 9.1 (1.2) years. Thus, both soy-based infant formula and breast milk are associated with an earlier age of allergies in FXS.

I did not observe a higher prevalence of material autoimmune disorders in the breastfeeding mothers [Q38: thyroid disorder: 16% no BM (*n* = 43), 13% BM (*n* = 139), 9.8% BM only (*n* = 41); autoimmune disorder: 14% no BM (*n* = 42); 7.9% BM (*n* = 139), 2.4% BM only (*n* = 41)]. Thus, the earlier age of onset of GI problems and allergies in FXS infants fed breast milk was not associated with reported autoimmune disorders in the mothers.

### 3.7. Comorbid Conditions as a Function of Exclusive Feeding with One Type of Milk

There was a limited number of participants reporting exclusive use of one type of milk. There were no statistically significant differences in the prevalence of autism, food allergies, GI problems, seizures or allergies (Appendix A). However, if the data is parsed based on sex, there were increased GI problems and allergies with cow milk formula compared to breast milk in males with a trend for increased autism (Table 9). There were too few females reporting exclusive use of one type of formula in the study population for a meaningful analysis. This analysis does not take into consideration the length of time on the indicated milk or initiation of solid foods. The average age of commencement of GI problems and allergies were significantly earlier with exclusive breast milk versus cow milk formula (Table 10). Overall, the findings indicate that breast milk is associated with protective effects against FXS comorbidities compared to formula use; however, in those children developing GI problems and allergies, those comorbidities occur significantly earlier with breast milk.

## 4. Discussion

There is a dearth of studies regarding how breastfeeding affects neurological development in FXS. To fill this gap, I evaluated retrospective survey data of participants enrolled in FORWARD to assess the impact of breast milk on disease outcomes in children with FXS. Infants with FXS often have poor latch and suck, making it more difficult to breastfeed [11]; thus, it was surprising that similar breastfeeding rates were found in the FORWARD participants in comparison to national averages [20]. Decreased prevalence of autism in FXS was also found in response to long-term or exclusive breastfeeding compared to formula feeding, but an earlier onset of GI problems and allergies. 

Health benefits associated with breast milk could be conferred through the higher bioavailability of nutrients, an altered gut microbiome, greater immunity from maternal antibodies, increased mother/child bonding, and/or better pre- and post-natal healthcare. For example, the decreased prevalence of autism in FXS as a function of breastfeeding could be due to insulin-like growth factor-1 (IGF-1) levels in breast milk. Axons in the autistic brain have a thinner coating of myelin [23]. IGF-1 regulates oligodendrocyte differentiation and subsequent myelin production [24]. Children with autism exhibit lower quantities of IGF-1 in cerebrospinal fluid and urine [25,26,27]. Thus, it has been hypothesized that an inadequate supply of IGF-1 results in abnormal myelination of brain neurons leading to the development of autism [28,29]. Delayed myelination is observed in *Fmr1^KO^* mice [30,31], and genetic reduction of insulin-like growth factor receptor-1 (IGF-1R) or treatment with a synthetic analog of a naturally occurring neurotropic peptide derived from IGF-1 corrects a multitude of disease phenotypes in *Fmr1^KO^* mice [32,33]. Breast milk has a higher quantity of human IGF-1 than bovine milk or infant formula [34] and is associated with elevated IGF-1 in preterm infants [35] and altered white matter volume in boys [36].

GI problems are common in children and adults with FXS [37,38]. Breast milk is protective against GI problems in infants at high risk for autism [39]; and likewise, this study found that breast milk was associated with the reduced prevalence of GI problems in males with FXS. Breast milk contains bioactive components including bacteria and antibodies that help to establish a healthy intestinal microbiota in infants. The intestinal microbiota of an exclusively breastfed baby is dominated by several species of *Bifidobacteria*, which are associated with reduced risk of infection in infancy, as well as a reduced risk of certain chronic illnesses in adulthood [40]. Formula feeding causes a shift in the gut bacterial profile [41], which could affect maturation of the epithelial cell barrier and gut permeability leading to allergic reactions [42].

There are racial and economic disparities in breastfeeding with higher rates among white, college-educated, middle class women [20]. This study did not collect data on socioeconomic indicators. It is likely that access to better pre- and post-natal health care contributes to improved health outcomes associated with breastfeeding. Another potentially confounding issue is that infants who are later diagnosed with autism may have dysregulated breastfeeding behaviors that resulted in a shorter duration of breastfeeding.

An important question raised by this study is whether breast milk is protective, or formula is detrimental, in regard to FXS comorbidities. I hypothesize that formula is detrimental in terms of autism in FXS because breast milk feeding (47% autism comorbidity) is more similar to no breast milk feeding (53%) than to exclusive breast milk (28%). Likewise, GI problems are more similar when comparing breast milk feeding (31% GI problem comorbidity) to no breast milk (33%) than to exclusive breast milk (23%) in FXS. Retrospective studies are considered a low level of evidence and prospective evaluation is required to justify specific dietary recommendations in FXS.

This study also evokes the question of whether or not there are differences in mothering habits or in breast milk from FXS premutation and full-mutation carriers that could contribute to disease severity in their children with FXS. For example, further studies are needed to explain the reported earlier onset of GI problems and allergies in response to breast milk in FXS. This study did not collect data on maternal premutation and full-mutation status; however, since participant eligibility included full-mutation FXS status, the biological mothers were all pre- or full-mutation carriers. Female premutation carriers are at higher risk for obsessive-compulsive symptoms and autoimmune disorders [43,44,45]. It is possible that those who breast fed were more attuned to observing GI and allergic symptoms in their children with FXS. Alternatively, many factors have a role in the development of allergies including the environment, maternal diet during pregnancy and lactation, timing of the introduction and the quality of solid foods, and the balance of immunologic factors in breast milk. The mothers who breastfed could have been more vigilant about maintaining germ-free conditions such that their babies had less exposure to germs and did not build tolerance to allergens. Or the earlier onset of GI issues and allergies associated with breast milk could be due to an increased use of antibiotics. 

It is unlikely that increased autoimmune antibodies in the breast milk caused the reported earlier onset of GI problems and allergies. Although higher titers of anti-neuronal antibodies have been reported in both autism and FXS [46,47,48], forty-five percent of female premutation carriers have at least one immune-mediated disorder [45] with autoimmune disease associated with seizures in their children with FXS [49], and antibodies are transmitted through breast milk, a higher prevalence of material autoimmune disorders in the breastfeeding mothers was not observed in this study. 

The nutritional profile of premutation and full-mutation breast milk remains to be determined. The most common substance in breast milk found to cause allergies is casein protein. However, casein protein is not the cause of earlier onset of allergies in the FXS breast milk cohort, as those participants were compared to participants with FXS fed cow milk formula. The scientific literature is inconclusive as regards the effects of breastfeeding in preventing or delaying the onset of specific food allergies [7,36,50]. This study did not find significant differences in food allergies dependent on breast milk although there was a trend for increased food allergies with soy-based infant formula [19].

The number of trinucleotide cytosine-guanine-guanine (CGG) repeats in premutation carriers correlates with multiple plasma metabolite levels including lower levels of fatty acids and the ketone body 3-hydroxybutyrate, as well as the ratio of *n*-3 to *n*-6 fatty acids [51]. Omega-3 fatty acids are essential for brain development, and a high fat ketogenic diet is highly effective in reducing seizures and hyperactivity in *Fmr1^KO^* mice [52]. *Fmr1^KO^* mice exhibit reduced triglycerides, total cholesterol, carnitine, leptin and insulin, and increased levels of free fatty acid and the ketone bodies acetone and acetoacetate [53]. Male participants with FXS exhibit low levels of total cholesterol, low-density lipoprotein (LDL) and high-density lipoprotein (HDL) [54]. In total, these data suggest altered lipid metabolism in both pre- and full-mutation FXS models. Considering the pivotal role cholesterol plays in myelination of the central nervous system [55], future research is warranted to test possible differences in the constituency of breast milk or lipid metabolism, as a function of *FMR1* mutation size, that could affect developmental outcomes in FXS. In addition, while breast milk from premutation carriers (*n* = 5) and controls (*n* = 25) are similar in terms of protein and lactose concentrations, the incidences of low protein and low lactose are higher in premutation carriers, while the quantity of the micronutrient zinc is lower [56]. Zinc is associated with the protein fraction of infant formulas, and the percentage of zinc in the soluble protein fraction in soy-based infant formula is very low compared to cow milk-based formula [57]. Low zinc levels may contribute to the development of autism through abnormal function of the zinc-metalloproteinase-brain-derived neurotrophic factor (BDNF) axis [58]. Macro- and micro-nutrient signatures of FXS premutation and full-mutation breast milk remain to be determined.

The limitations of this study include the absence of longitudinal studies, which are not possible in this population because most participants are not diagnosed until three years of age or older after breastfeeding has been discontinued, recall and reporting biases which are potential issues with all retrospective studies, and unknown CGG repeat number in the *FMR1* gene and socioeconomic status of the mothers. Recall bias, particularly the use of breast milk, was not expected to be a problem as parents typically have specific reasons for their choice of infant feeding and switch for specific reasons such as GI problems or allergies. Reporting bias could be an issue with the reported age of onset for GI problems and allergies because mothers are solely responsible for breastfeeding whereas fathers, who may be more risk tolerant, can assist with formula feeding and due to the timing between the cessation of breastfeeding and the survey. It should be noted that there are practical implications of exclusive, prolonged breastfeeding. The major reasons reported for discontinuation of breast milk feeding in this study were inadequate quantify of milk; child age and transition to baby food and sippy cup; and mother need to wean child for work, health or convenience-related issues. Also of note, the average age of an FXS diagnosis is 35–37 months in boys and 41 months in girls with an average of 10 symptom-related visits to health care professionals before a diagnostic FXS DNA test is ordered [59]. An early nutritional intervention that reduced the severity of FXS phenotypes, such as supplementation of breast milk with lipids, would support including FXS in the newborn screening (NBS) panel. 

The strengths of this study include the survey links parent-reported data on breastfeeding to the development of comorbid conditions including autism in their children with FXS and the higher than expected rate of breastfeeding in this population allowed evaluation of the timing of GI and allergy comorbidities. Although clinical autism diagnostic test scores were not available, it is reassuring that caregiver-reported incidence of autism concurs with survey questions adapted from standardized instruments that assess cognitive ability and autistic behaviors (i.e., Ages and Stages Questionnaire for 36-months, Modified Checklist for Autism in Toddlers-Revised, and the Social Responsiveness Scale parental survey). Of 32 questions on our survey, the caregiver-reported autism cohort exhibited statistically significant differences with Bonferroni correction on 14 questions and statistically significant differences without Bonferroni correction on 24 questions. All differences in outcomes were in the expected direction for an autism diagnosis.

Collectively, the association between breastmilk and decreased comorbidities in FXS concurs with extensive literature supporting the opinion that, “Human breastmilk is therefore not only a perfectly adapted nutritional supply for the infant, but probably the most specific personalised medicine he or she is likely to receive, given at a time when gene expression is being fine-tuned for life.” [60]. It remains to be determined if there are macro- or micronutrient deficiencies in breastmilk from pre- or full-mutation carriers that could be supplemented to add further benefit.

## 5. Conclusions

In conclusion, this retrospective survey study provides data regarding an association between breastfeeding and the reduced prevalence of autism in participants with FXS recruited through FORWARD. Although the overall prevalence of GI problems and allergies were not statistically different with breast milk in the total study population and were higher with cow milk formula compared to breast milk in males, the age of onset of GI problems and allergies in FXS participants were significantly younger in those fed breast milk. Overall, these findings indicate that breastmilk is associated with improved outcomes in FXS compared to infant formula and that mothers should be encouraged to breastfeed. The findings also support further studies to identify potentially altered levels of breastmilk components as a function of maternal *FMR1* mutation status, which may affect development in their children. FXS is typically not diagnosed until 3 years of age or older; however, NBS protocols are validated and could be implemented if an early-life intervention such as a dietary supplement were available.

## Figures and Tables

**Table 1 nutrients-13-01785-t001:** Timeframe for any breastfeeding in FXS study population.

Timeframe	Total Population BM ^a^ % (*n* = 141)	Males BM % (*n* = 106)	Females BM % (*n* = 35)	Males vs. Females *p* ^b^
2 weeks or less	5.7	5.7	5.7	1.0 ^c^
3–10 weeks	13	12	17	0.46
at least 3 months	81	82	71	0.18
at least 6 months	73	75	60	0.078
12 months or longer	41	41	37	0.72
only BM ^d^	29	30	26	0.61

^a^ BM = breast milk. ^b^ Chi-squared tests were used unless any variable contained less than *n* = 5, in which case Fisher exact test was used. ^c^ Fisher exact test. ^d^ Two participants were excluded from the BM only cohort even though Questions #23 (Was your child with Fragile X Syndrome fed any cow milk formula in his or her first year of life?), #28 (Was your child with Fragile X Syndrome fed any soy-based formula in his or her first year of life?) and #33 (Some examples of specialty formulas include amino acid, rice, or meat-based formulas. Was your child with Fragile X Syndrome fed any specialty formulas in his or her first year of life?) were answered “No” because Question #22 (Some examples of why people might choose to stop feeding their child breastmilk include being painful for the mother, or the baby not getting enough milk What was the reason your child stopped being fed breastmilk?) indicated that formula was used.

**Table 2 nutrients-13-01785-t002:** Analysis of FXS comorbidities as a function of any breast milk.

Phenotype	BM ^a^ % (*n*)	No BM % (*n*)	*p* ^b^	OR	95% CI
none ^d^	22 (134)	19 (52)	0.56	0.79	0.36–1.8
autism	47 (131)	53 (47)	0.49	0.79	0.41–1.5
food allergies	11 (130)	11 (47)	0.98	1.0	0.34–3.0
diabetes	0.75 (134)	2.1 (48)	0.46 ^c^	0.35	0.022–5.8
GI problems	31 (137)	33 (51)	0.73	0.88	0.45–1.8
seizures	12 (139)	17 (52)	0.36	0.67	0.28–1.6
allergies	36 (138)	42 (52)	0.39	0.75	0.39–1.4

^a^ BM = breast milk. ^b^ Chi-squared tests were used unless any variable contained less than *n* = 5, in which case Fisher exact test was used. ^c^ Fisher exact test. ^d^ none is defined as no reporting of autism, food, allergies, diabetes, GI problems, seizures or allergies.

**Table 3 nutrients-13-01785-t003:** Analysis of FXS comorbidities as a function of duration of breast milk.

Phenotype	No BM ^a^ % (*n*)	BM 3 mo % (*n*)	BM 6 mo % (*n*)	BM 12 mo % (*n*)	BM 3 mo *p* ^b^, OR, 95% CI	BM 6 mo *p* ^b^, OR, 95% CI	BM 12 mo *p* ^b^, OR, 95% CI
none ^d^	19 (52)	25 (110)	26 (99)	33 (56)	0.38, 0.70, 0.31–1.6	0.27, 0.63, 0.28–1.4	0.085, 0.46, 0.19–1.1
autism	53 (47)	46 (106)	44 (96)	32 (56)	0.43, 0.76, 0.38–1.5	0.29, 0.68, 0.34–1.4	0.031, 0.42, 0.19–0.93
food allergies	11 (47)	12 (104)	13 (96)	11 (53)	0.87, 1.1, 0.36–3.3	0.75, 1.2, 0.40–3.6	0.91, 1.1, 0.31–3.8
diabetes	2.1 (48)	0.93 (108)	1.0 (98)	0 (55)	0.52 ^c^, 0.44, 0.027–7.2	0.55 ^c^, 0.48, 0.030–7.9	0.47 ^c^, n/a, n/a
GI problems	33 (51)	29 (112)	25 (102)	25 (57)	0.54, 0.80, 0.39–1.6	0.31, 0.68, 0.33–1.4	0.31, 0.65, 0.28–1.5
seizures	17 (52)	14 (111)	12 (100)	5.2 (58)	0.52, 0.75, 0.30–1.8	0.37, 0.65, 0.26–1.7	0.064 ^c^, 0.26, 0.066–1.0
allergies	42 (52)	36 (110)	37 (100)	34 (56)	0.47, 0.78, 0.40–1.5	0.52, 0.80, 0.40–1.6	0.37, 0.70, 0.32–1.5

^a^ BM = breast milk. ^b^ Chi-squared tests were used unless any variable contained less than *n* = 5, in which case Fisher exact test was used. Data were compared with no BM data. ^c^ Fisher exact test. ^d^ None is defined as no reporting of autism, food, allergies, diabetes, GI problems, seizures or allergies.

**Table 4 nutrients-13-01785-t004:** Analysis of caregiver-reported autism and autism phenotypes.

	Q1a (autism, “YES”)	Q1a (autism, “NO”)	Fold Change	*p* ^a^	Significant ^a^
Q40a (talk) (average, SEM, N)	2.2 (0.12) (85)	3.0 (0.13) (88)	0.73	5.2 × 10^−6^	Yes
Q40b (say name)	2.1 (0.12) (87)	2.7 (0.15) (88)	0.77	2.0 × 10^−3^	No
Q40c (respond)	3.2 (0.11) (87)	4.1 (0.090) (88)	0.78	3.7 × 10^−9^	Yes
Q40d (3 word sentences)	1.7 (0.10) (88)	2.4 (0.15) (88)	0.71	3.0 × 10^4^	Yes
Q40e (words to request things)	2.1 (0.12) (88)	2.8 (0.14) (87)	0.77	6.8 × 10^−4^	Yes
Q41a (correctly identify)	2.7 (0.13) (86)	3.3 (0.13, 86)	0.81	1.2 × 10^−3^	Yes
Q41b (follow directions)	2.6 (0.11) (87)	3.3 (0.11) (88)	0.79	1.9–5	Yes
Q41c (point to request)	2.8 (0.13) (87)	3.6 (0.11) (88)	0.76	7.2 × 10^−7^	Yes
Q41d (copy others)	2.7 (0.12) (87)	3.6 (0.10) (87)	0.76	1.5 × 10^−7^	Yes
Q41e (play pretend)	1.6 (0.11) (87)	2.4 (0.14) (87)	0.67	1.4 × 10^−5^	Yes
Q41f (have savant ability)	1.4 (0.10) (87)	1.3 (0.078) (88)	1.09	0.34	No
Q43a (likes motion)	4.1 (0.11) (86)	4.0 (0.098) (88)	1.02	0.64	No
Q43b (walk)	4.1 (0.10) (86)	4.3 (0.093) (87)	0.95	0.16	No
Q43c (toe walk)	2.4 (0.14) (85)	2.0 (0.15) (87)	1.2	0.083	No
Q43d (picks up small objects)	3.5 (0.11) (87)	3.8 (0.12) (87)	0.92	0.063	No
Q43e (feeds self with spoon)	2.6 (0.13) (87)	3.6 (0.12) (87)	0.74	1.4 × 10^−6^	Yes
Q43f (help dress self)	2.3 (0.12) (87)	2.9 (0.13) (88)	0.79	7.3 × 10^−4^	Yes
Q44a (upset by loud noises)	3.4 (0.15) (86)	2.9 (0.13) (88)	1.16	0.017	No
Q44b (rocking, hand flapping)	3.8 (0.14) (87)	3.0 (0.17) (88)	1.26	4.5 × 10^−4^	Yes
Q44c (cry excessively)	2.4 (0.13) (87)	2.4 (0.13) (88)	1.03	0.74	No
Q44d (temper outbursts)	3.1 (0.13) (87)	3.0 (0.13) (88)	1.06	0.31	No
Q44e (isolate self)	2.7 (0.12) (87)	2.3 (0.13) (87)	1.19	0.017	No
Q44f (injure self)	2.0 (0.14) (87)	1.6 (0.13) (88)	1.23	0.047	No
Q45a (upset by minor changes)	3.5 (0.11) (87)	3.1 (0.12) (88)	1.13	0.013	No
Q45b (difficulty expressing needs)	4.2 (0.087) (87)	3.8 (0.11) (88)	1.09	0.013	No
Q45c (hate crowds)	3.8 (0.13) (87)	3.1 (0.13) (88)	1.21	5.7 × 10^−4^	Yes
Q45d (not liked to be touched)	2.5 (0.11) (87)	2.2 (0.13) (88)	1.16	0.050	No
Q45e (like to play with other kids)	2.4 (0.12) (87)	3.2 (0.11) (88)	0.76	3.7 × 10^−6^	Yes
Q46a (anxiety problem)	3.7 (0.14) (87)	3.2 (0.15) (88)	1.17	0.010	No
Q46b (hearing problem)	2.0 (0.14) (87)	1.6 (0.12) (88)	1.31	7.8 × 10^−3^	No
Q46c (vision problem)	1.8 (0.12) (87)	1.5 (0.12) (88)	1.22	0.067	No
Q46d (learning problem)	4.6 (0.079) (87)	4.2 (0.12) (88)	1.09	0.012	No

^a^ The result is statistically significant after applying a Bonferroni correction for 32 questions such that statistical significance is defined as *p* < 0.0016.

**Table 5 nutrients-13-01785-t005:** Linear regression analysis of FXS comorbidities as a function of time on breast milk.

Phenotype	Regression Coefficient ^a^	Correlation Coefficient
none ^b^	1.1	0.96
autism	−1.7	0.98
food allergies	−0.0095	0.0026
diabetes	−0.16	0.88
GI problems	−0.65	0.75
seizures	−0.97	0.99
allergies	−0.56	0.72

^a^ The Regression Coefficient is the constant (a) from the regression line y = ax + b that represents the rate of change of comorbidity prevalence (y) as a function of time on breast milk (x). ^b^ none is defined as the absence of autism, food, allergies, diabetes, GI problems, seizures and allergies.

**Table 6 nutrients-13-01785-t006:** Analysis of FXS comorbidities as a function of only breast milk.

Phenotype	No BM ^a^ % (N)	BM only % (*n*)	*p* ^b^	OR	95% CI
none ^d^	19 (52)	38 (40)	0.051	0.40	0.16–1.0
autism ^e^	53 (47)	28 (39)	0.019	0.35	0.14–0.85
food allergies	11 (47)	13 (38)	0.72	1.3	0.34–4.8
diabetes	2.1 (48)	0 (38)	1.0 ^c^	n/a	n/a
GI problems	33 (51)	23 (40)	0.26	0.58	0.23–1.5
Seizures	17 (52)	7.3 (41)	0.22 ^c^	0.38	0.095–1.5
allergies	42 (52)	31 (39)	0.26	0.61	0.25–1.5

^a^ BM = breast milk. ^b^ Chi-squared tests were used unless any variable contained less than *n* = 5, in which case Fisher exact test was used. Data were compared with no BM data. ^c^ Fisher exact test. ^d^ none is defined as no reporting of autism, food, allergies, diabetes, GI problems, seizures or allergies. ^e^ Two participants were excluded from the BM only cohort even though Questions #23 (Was your child with Fragile X Syndrome fed any cow milk formula in his or her first year of life?), #28 (Was your child with Fragile X Syndrome fed any soy-based formula in his or her first year of life?) and #33 (Some examples of specialty formulas include amino acid, rice, or meat-based formulas. Was your child with Fragile X Syndrome fed any specialty formulas in his or her first year of life?) were answered “No” because Question #22 (Some examples of why people might choose to stop feeding their child breastmilk include being painful for the mother, or the baby not getting enough milk What was the reason your child stopped being fed breastmilk?) indicated that formula was used.

**Table 7 nutrients-13-01785-t007:** Reasons children with FXS were fed breast milk or stopped being fed breast milk.

***n***	**Reasons to Start ^a^**
123	Healthiest for the baby (believed best for baby, best for development, strong immunity/antigen benefit, most natural)
16	Parental choice/belief/instinct
11	Doctor recommended
8	Family recommended/custom
7	To bond with baby
2	To save money
***n***	**Reasons to Stop ^b^**
47	Not enough milk
46	Child age-related (time to stop, transitioned to baby food, using a sippy cup)
31	Mother wanted/needed to wean child (work-related, health of mother-related, eat dairy again, convenience, time)
18	Painful to mother
16	Child health-related problem (GERD, latching issue, poor suck, rejecting breast, missed developmental milestones)
6	Lack of interest by child
4	Mother became pregnant or wanted to conceive again

^a^ A total of 142 participants provided responses. Many participants provided multiple reasons. ^b^ A total of 144 participants provided responses. Many participants provided multiple reasons.

**Table 8 nutrients-13-01785-t008:** Timeframe of GI problems and allergies in FXS study population as a function of breast milk.

**Age of Onset of GI Problems**	**BM ^a^ % (*n* = 39)**	**No BM % (*n* = 17)**	***p***	**OR**	**95% CI**
0–2 weeks (%)	33	41	0.57 ^b^	0.71	0.22–2.3
0–12 months (%)	69	65	0.74 ^b^	1.2	0.37–4.1
0–3 years (%)	95	65	0.007 ^c^	10.1	1.8–57
Mean age in months (SEM)	22 (9.1)	65 (24)	0.044 ^d^	n/a	n/a
**Age of Allergy Onset**	**BM % (*n* = 45)**	**No BM % (*n* = 19)**	***p***	**OR**	**95% CI**
0–2 weeks (%)	4.4	11	0.58 ^c^	0.40	0.015–3.0
0–12 months (%)	33	26	0.58 ^b^	1.4	0.42–4.6
0–3 years (%)	78	47	0.016 ^b^	3.9	1.2–12
Mean age in months (SEM)	29 (3.9)	67 (13)	0.00054 ^d^	n/a	n/a

^a^ BM = breast milk. ^b^ Chi-squared test. ^c^ Fisher exact test. ^d^ Student *t*-test.

**Table 9 nutrients-13-01785-t009:** Analysis of FXS comorbidities as a function of exclusive milk use in males.

Phenotype	BM % (*n*)	CM % (*n*)	SM % (*n*)	*p* ^a^	*p* ^b^	*p* ^c^	*p* ^d^	OR	95% CI
autism	27 (30)	67 (9)	50 (4)	0.070	0.048	0.56	0.61	5.5 ^e^	1.1–27 ^e^
food allergies	10 (30)	14 (7)	67 (3)	0.071	1.0	0.053	0.18	18 ^f^	1.2–263 ^f^
GI problems	19 (31)	67 (9)	25 (4)	0.026	0.012	1.0	0.27	8.3 ^e^	1.6–43 ^e^
seizures	9.4 (32)	0 (9)	25 (4)	0.46	1.0	0.39	0.31	n/a	n/a
allergies	32 (31)	78 (9)	50 (4)	0.045	0.023	0.59	0.53	7.4 ^e^	1.3–42 ^e^

^a^ Fisher exact test (3 × 2 contingency table) was used to compare breast milk (BM), cow milk formula (CM) and soy-based infant formula (SM). ^b^ Fisher exact test (2 × 2 contingency table) was used to compare BM and CM. ^c^ Fisher exact test (2 × 2 contingency table) was used to compare BM and SM. ^d^ Fisher exact test (2 × 2 contingency table) was used to compare CM and SM. ^e^ CM versus BM. ^f^ SM versus BM.

**Table 10 nutrients-13-01785-t010:** Age of commencement of GI problems and allergies in FXS study population as a function of exclusive breast milk versus cow milk formula.

Comorbidity	BM ^a^	CM ^b^	*p* ^c^
GI problems, Mean age in years (SEM) (*n*)	0.90 (0.55) (5)	12 (3.4) (6)	0.018
Allergies, Mean age in years (SEM) (*n*)	1.8 (0.51) (10)	9.4 (1.6) (7)	<0.0001

^a^ BM = breast milk. ^b^ CM = cow milk formula. ^c^ Student *t*-test.

## Data Availability

Data is contained within the article or Appendix A.

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
