# Peer review of "Consumption of Breast Milk Is Associated with Decreased Prevalence of Autism in Fragile X Syndrome"

_nutrients, 2021, doi:10.3390/nu13061785_

Round 1
Reviewer 1 Report
I agree with the author that breastfeeding in theory should be beneficial but I don't think one sentence in the introduction makes the case sufficiently.
I agree that important benefits of breastfeeding have been uncovered by this research but wonder how this can be applicable to those diagnosed with Fragile X because they are often not diagnosed until age 3. This and much more research* confirms that all babies should be breastfed (or breastmilk fed if unable to feed at the breast). So the broad recommendation would be that all babies should be breastfed with this research providing further evidence for that recommendation.
I am not aware of genetic changes in breastmilk, so am unsure of this validity of this statement:
'These findings support future examination of altered levels of potential bio- active components in breast milk as a function of maternal FMR1 mutation status and the potential effects on development in their children with FXS.'
It would be the lack of breastmilk that has lead to poorer outcomes in those with Fragile X that have been formula fed. If we want to encourage mothers to breastfeed, we don't want to undermine their confidence in their breastmilk by suggesting it may be 'deficient'. Any breastmilk is better than formula.
*Victora, C. G., Bahl, R., Barros, A. J., França, G. V., Horton, S., Krasevec, J., ... & Group, T. L. B. S. (2016). Breastfeeding in the 21st century: epidemiology, mechanisms, and lifelong effect. The Lancet, 387(10017), 475-490.
Author Response
Response to Reviewers
Thank you for the careful review of my work. Please find my responses below.
Review #1:
I agree with the author that breastfeeding in theory should be beneficial but I don't think one sentence in the introduction makes the case sufficiently. I have added a paragraph to the Introduction summarizing the benefits of breastfeeding:
The American Academy of Pediatrics recommends exclusive breastfeeding for the first 6 months of life and continued breastfeeding with the introduction of complementary foods [1]. Breastfeeding is the ideal nutrition for babies and is associated with numerous health benefits, for example, reduced incidence of autism spectrum disorder (ASD), attention-deficit/hyperactivity (ADHD) disorder, obesity, sudden infant death syndrome (SIDS), infections and atopic disease. Specifically, children with ASD, whether by clinical diagnosis or self-report, are significantly less likely to have been breastfed [2]. Data from the Centers for Disease Control and Prevention (CDC) regarding several million children in the United States indicate a highly significant inverse relationship between ADHD and exclusive 3-month and 6-month breastfeeding [3]. Meta-analysis with a total over 226,000 participants indicates that breastfeeding is associated with significantly reduced risk of obesity in children with a dose response effect dependent on duration of breastfeeding [4]. Exclusive breastfeeding for less than 4 months is associated with increased risk of chest infection and diarrhea [5]. Any breastfeeding greater than 2 months is protective against sudden infant death syndrome (SIDS) with greater protection observed with longer duration of breastfeeding [6]. And exclusive breastfeeding for 3-4 months decreases the incidence of eczema while longer breastfeeding is protective against wheezing in the first 2 years of life [7].
I agree that important benefits of breastfeeding have been uncovered by this research but wonder how this can be applicable to those diagnosed with Fragile X because they are often not diagnosed until age 3. This and much more research* confirms that all babies should be breastfed (or breastmilk fed if unable to feed at the breast). So the broad recommendation would be that all babies should be breastfed with this research providing further evidence for that recommendation. The following statements have been added to the Results and Conclusions, respectively:
Collectively, the association between breastmilk and decreased comorbidities in FXS concurs with extensive literature supporting the opinion that, “Human breastmilk is therefore not only a perfectly adapted nutritional supply for the infant, but probably the most specific personalised medicine he or she is likely to receive, given at a time when gene expression is being fine-tuned for life.” [60]. It remains to be determined if there are macro- or micronutrient deficiencies in breastmilk from pre- or full-mutation carriers that could be supplemented to add further benefit.
Overall, these findings indicate that breastmilk is associated with improved outcomes in FXS compared to infant formula and that mothers should be encouraged to breastfeed. The findings also support further studies to identify potentially altered levels of breastmilk components as a function of maternal FMR1 mutation status, which may affect development in their children. FXS is typically not diagnosed until 3 years of age or older; however, NBS protocols are validated and could be implemented if an early-life intervention such as a dietary supplement were available.
I am not aware of genetic changes in breastmilk, so am unsure of this validity of this statement:
'These findings support future examination of altered levels of potential bio- active components in breast milk as a function of maternal FMR1 mutation status and the potential effects on development in their children with FXS.' There is published work from the Hagerman laboratory showing a higher incidence of low protein and low lactose as well as lower levels of zinc in breast milk from premutation carriers (lines 381-382). Although not a large study, these findings warrant further investigation of the potential differences in premutation breast milk and effects on development in their children. I revised the conclusion statements (described above) and deleted the phrase “bioactive” to avoid undermining confidence in breastmilk.
It would be the lack of breastmilk that has lead to poorer outcomes in those with Fragile X that have been formula fed. If we want to encourage mothers to breastfeed, we don't want to undermine their confidence in their breastmilk by suggesting it may be 'deficient'. Any breastmilk is better than formula. I agree that breastmilk is better than formula and we want to encourage mothers to breastfeed. A statement to this effect has been added to the Conclusions (see above).
*Victora, C. G., Bahl, R., Barros, A. J., França, G. V., Horton, S., Krasevec, J., ... & Group, T. L. B. S. (2016). Breastfeeding in the 21st century: epidemiology, mechanisms, and lifelong effect. The Lancet, 387(10017), 475-490.
This citation has been added to the manuscript as reference #60.
Reviewer 2 Report
In this interesting manscript, Dr Westmark evaluate the associations between the consumption of breast milk during infancy and the prevalence of autism, allergies, diabetes, gastrointestinal (GI) problems and seizures in FXS. The study design is retrospective in nature with data collected through a specific questionnaire. The author concludes that that long
term or exclusive use of breast milk is associated with reduced prevalence of "autism" and other key comorbidities in FXS, although breast milk is associated with the earlier development of GI problems and allergies.
The study is well conducted, the methodology and the statistic analysis are overall adequate .
I have only a major concern.
The main problem of this work is the absence of standardized tools to diagnose "autism". Autism is a complex definition which is much better represented in the form of a clinical spectrum than as a single category. As Fragile X syndrome is commonly associated with complex autistic-like behaviour, the severity of symptoms in a patient could be emphasized or diminished by his attending clinician in a survey.
Due to ther relevance of this problem, to derive secure conclusions, it would be necessary to state which standardized tests were used in each patient to confirm the diagnosis of autism (or, better, of autism spectrum disorder) and to report the score of these tests .
Author Response
Response to Reviewers
Thank you for the careful review of my work. Please find my responses below.
Review #2:
In this interesting manscript, Dr Westmark evaluate the associations between the consumption of breast milk during infancy and the prevalence of autism, allergies, diabetes, gastrointestinal (GI) problems and seizures in FXS. The study design is retrospective in nature with data collected through a specific questionnaire. The author concludes that that long
term or exclusive use of breast milk is associated with reduced prevalence of "autism" and other key comorbidities in FXS, although breast milk is associated with the earlier development of GI problems and allergies.
The study is well conducted, the methodology and the statistic analysis are overall adequate.
I have only a major concern.
The main problem of this work is the absence of standardized tools to diagnose "autism". Autism is a complex definition which is much better represented in the form of a clinical spectrum than as a single category. As Fragile X syndrome is commonly associated with complex autistic-like behaviour, the severity of symptoms in a patient could be emphasized or diminished by his attending clinician in a survey. Due to the relevance of this problem, to derive secure conclusions, it would be necessary to state which standardized tests were used in each patient to confirm the diagnosis of autism (or, better, of autism spectrum disorder) and to report the score of these tests.
I agree that scores from standardized autism tools would provide a higher degree of evidence for the study. Unfortunately, clinical diagnostic scores for autism were not available for the subjects because fragile X is a rare disorder and the FORWARD IRB protocol prohibited providing medical record data at the individual levels to researchers. Thus, for this retrospective survey, all of the data including the autism diagnosis had to be collected from the survey and is caregiver-reported. Our goal is to use this retrospective, caregiver-reported data to justify prospective evaluation of infant feeding and autism in fragile X infants in collaboration with a recently initiated newborn screening program in North Carolina where infants with fragile X are identified at birth and will be evaluated with standardized autism diagnostic tools.
To mitigate for the lack of clinical autism diagnostic scores in this study, I co-submitted the manuscript entitled, “Exploratory Associations Between Infant Feeding and Autistic Behaviors in Fragile X Syndrome”, which examined the prevalence of 32 individual autistic behaviors in fragile X as a function of infant diet using the same study population/survey. Standardized tests could not be included in the survey due to copyright issues and the inability to make modifications to verb tense, etc. for a retrospective survey. However, the 32 questions regarding autistic behaviors were derived from standardized tools (Ages and Stages Questionnaire for 36-months, Modified Checklist for Autism in Toddlers-Revised, Social Responsiveness Scale parental survey).
Again, I agree that standardized tests are optimal, but with a rare disorder where infants are not identified until 3 years of age, I had to conduct a retrospective study. The FORWARD registry was the best available to recruit fragile X subjects and it prohibited sharing medical record data at the individual level.